# (LA)yer-neigh(BOR) Sampling: Defusing Neighborhood Explosion in GNNs

## Abstract

Graph Neural Networks have recently received a significant attention, however, training them at a large scale still remains a challenge. Minibatch training coupled with sampling is used to alleviate this challenge. Even so existing approaches either suffer from the neighborhood explosion phenomenon or do not have good performance. To deal with these issues, we propose a new sampling algorithm called LAyer-neighBOR sampling (LABOR). It is designed to be a direct replacement for Neighbor Sampling with the same fanout hyperparameter while sampling upto $7\times$ fewer vertices, without sacrificing quality. By design, the variance of the estimator of each vertex matches Neighbor Sampling from the point of view of a single vertex. Moreover under the same vertex sampling budget constraints, LABOR converges faster than existing layer sampling approaches and can use upto $112\times$ larger batch size compared to Neighbor Sampling.

## 1 Introduction

Graph Neural Networks (GNN) Hamilton et al. (2017); Kipf & Welling (2017) have become de facto models for representation learning on graph structured data. Hence they have started being deployed in production systems Ying et al. (2018); Niu et al. (2020). These models iteratively update the node embeddings by passing messages along the direction of the edges in the given graph with nonlinearities in between different layers. With $l$ layers, the computed node embeddings contain information from the $l$-hop neighborhood of the seed vertex.

In the production setting, the GNN models need to be trained on billion-scale graphs (Ching et al., 2015; Ying et al., 2018). The training of these models takes hours to days even on distributed systems Zheng et al. (2022b;a). As in general Deep Neural Networks (DNN), it is more efficient to use minibatch training (Bertsekas, 1994) on GNNs, even though it is a bit trickier in this case. The node embeddings in GNNs depend recursively on their set of neighbors' embeddings, so when there are $l$ layers, this dependency spans the $l$-hop neighborhood of the node. Real world graphs usually have a very small diameter and if $l$ is large, the $l$-hop neighborhood may very well span the entire graph, also known as the Neighborhood Explosion Phenomenon (NEP) (Zeng et al., 2020).

To solve these issues, researchers proposed sampling a subgraph of the $l$-hop neighborhood of the nodes in the batch. There are mainly three different approaches: Node-based, Layer-based and Subgraph-based methods. Node-based sampling methods (Hamilton et al., 2017; Chen et al., 2018a; Liu et al., 2020; Zhang et al., 2021) sample independently and recursively for each node. It was noticed that node-based methods sample subgraphs that are too shallow, i.e., with a low ratio of number of edges to nodes. Thus layer-based sampling methods were proposed (Chen et al., 2018b; Zou et al., 2019; Huang et al., 2018; Dong et al., 2021), where the sampling for the whole layer is done collectively. On the other hand subgraph sampling methods (Chiang et al., 2019; Zeng et al., 2020; Hu et al., 2020b; Zeng et al., 2021) do not use the recursive layer by layer sampling scheme used in the node- and layer-based sampling methods and instead tend to use the same subgraph for all of the layers. Some of these sampling methods take the magnitudes of embeddings into account (Liu et al., 2020; Zhang et al., 2021; Huang et al., 2018), while others, such as Chen et al. (2018a); Cong et al. (2021), cache the historical embeddings to reduce the variance of the computed approximate embeddings. There are methods sampling from a vertex cache Dong et al. (2021) filled with popular vertices. Most of these approaches are orthogonal to each other and they can be incorporated into other sampling algorithms.

Node-based sampling methods suffer the most from the NEP but they guarantee a good approximation for each embedding by ensuring each vertex gets $k$ neighbors which is the only hyperparameter of the sampling algorithm. Layer-based sampling methods do not suffer as much from the NEP because number of vertices sampled is a hyperparameter but they can not guarantee that each vertex approximation is good enough and also their hyperparameters are hard to reason with, number of nodes to sample at each layer depends highly on the graph structure (as the numbers in Table 2 show). Subgraph sampling methods usually have more bias than their node- and layer-based counterparts. Hence, in this paper, we focus on the node- and layer-based sampling methods and combine their advantages. The major contributions of this work can be listed as follows:

- We propose a new sampling algorithm called LABOR, combining advantages of neighbor and layer sampling approaches using Poisson Sampling. LABOR correlates the sampling procedures of the given set of seed nodes so that the sampled vertices from different seeds have a lot of overlap, resulting into a $7\times$ reduction in computation, memory and communication. Furthermore, LABOR has the same hyperparameters as neighbor sampling to use as a drop-in replacement and can speed up training by upto $2.6\times$.
- We experimentally verify our findings, show that our proposed sampling algorithm LABOR outperforms both neighbor sampling and layer sampling approaches. LABOR can enjoy a batch-size of upto $112\times$ larger than NS while sampling the same number of vertices.

## 2 BACKGROUND

**Graph Neural Networks:** Given a directed graph $\mathcal{G} = (V, E)$, where $V$ and $E \subset V \times V$ are vertex and edge sets respectively, $(t \to s) \in E$ denotes an edge from a source vertex $t \in V$ to a destination vertex $s \in V$, and $A_{ts}$ denotes the corresponding edge weight if provided. If we have a batch of seed vertices $S \subset V$, let us define $l$-hop neighborhood $N^l(S)$ for the incoming edges as follows:

$$N(s) = \{t | (t \to s) \in E\}, \ N^1(S) = N(S) = \cup_{s \in S} N(s), \ N^l(S) = N(N^{l-1}(S)) \tag{1}$$

Let us also define the degree $d_s$ of vertex $s$ as $d_s = |N(s)|$. To simplify the discussion, let's assume uniform edge weights, $A_{ts} = 1, \forall (t \to s) \in E$. Then, our goal is to estimate the following for each vertex $s \in S$, where $H_t^{(l-1)}$ is defined as the embedding of the vertex $t$ at layer $l-1$, and $W^{(l-1)}$ is the trainable weight matrix at layer $l-1$, and $\sigma$ is the nonlinear activation function (Hamilton et al., 2017):

$$Z_s^{(l)} = \frac{1}{d_s} \sum_{t \to s} H_t^{(l-1)} W^{(l-1)}, \ H_s^{(l)} = \sigma(Z_s^{(l)}) \tag{2}$$

**Exact Stochastic Gradient Descent:** If we have a node prediction task and $V_t \subseteq V$ is the set of training vertices, $y_s, s \in V_t$ are the labels of the prediction task, and $\ell$ is the loss function for the prediction task, then our goal is to minimize the following loss function: $\frac{1}{|V_t|} \sum_{s \in V_t} \ell(y_s, Z_s^l)$. Replacing $V_t$ in the loss function with $S \subset V_t$ for each iteration of gradient descent, we get stochastic gradient descent for GNNs. However with $l$ layers, the computation dependency is on $N^l(S)$, which reaches large portion of the real world graphs, i.e. $|N^l(S)| \approx |V|$, making each iteration costly both in terms of computation and memory.

**Neighbor Sampling:** Neighbor sampling approach was proposed by Hamilton et al. (2017) to approximate $Z_s^{(l)}$ for each $s \in S$ with a subset of $N^l(S)$. Given a fanout hyperparameter $k$, this subset is computed recursively by randomly picking $k$ neighbors for each $s \in S$ from $N(s)$ to form the next layer $S^1$, that is a subset of $N^1(S)$. If $d_s \leq k$, then the exact neighborhood $N(s)$ is used. For the next layer, $S^1$ is treated as the new set of seed vertices and this procedure is applied recursively.

**Revisiting LADIES, Dependent Layer-based Sampling** From now on, we will drop the layer notation and focus on a single layer and also ignore the nonlinearities. Let us define $M_t = H_t W$ as a shorthand notation. Then our goal is to approximate:

$$H_s = \frac{1}{d_s} \sum_{t \to s} M_t \tag{3}$$

If we assign probabilities $\pi_t > 0, \forall t \in N(S)$ and normalize it so that $\sum_{t \in N(S)} \pi_t = 1$, then use sampling with replacement to sample $T \subset N(S)$ with $|T| = n$, where $n$ is the number of vertices to sample given as input to the LADIES algorithm and $T$ is a multiset possibly with multiple copies of the same vertices, and let $\tilde{d}_s = |T \cap N(s)|$ which is the number of sampled vertices for a given vertex $s$, we get the following two possible estimators for each vertex $s \in S$:

$$H'_s = \frac{1}{n d_s} \sum_{t \in T \cap N(s)} \frac{M_t}{\pi_t} \tag{4}$$

$$H''_s = \frac{\sum_{t \in T \cap N(s)} \frac{M_t}{\pi_t}}{\sum_{t \in T \cap N(s)} \frac{1}{\pi_t}} \tag{5}$$

Note that $H'_s$ in Eq. 4 is the Thompson-Horvitz estimator and the $H''_s$ in Eq. 5 is the Hajek estimator. For a comparison between the two and how to get an even better estimator by combining them, see Khan & Ugander (2021). The formulation in the LADIES paper uses $H'_s$, but it proposes to row-normalize the sampled adjacency matrix, meaning they use $H''_s$ in their implementation. However, analysing the variance of the Thompson-Horvitz estimator is simpler and its variance serves as an upper bound for the variance of the Hajek estimator when $|M_t|$ and $\pi_t$ are uncorrelated Khan & Ugander (2021); Dorfman (1997), which we assume to be true in our case.

$$\text{Var}(H''_s) \le \text{Var}(H'_s) = \frac{1}{\tilde{d}_s d_s^2} \sum_{t \to s} \pi_t \sum_{t' \to s} \frac{\text{Var}(M_{t'})}{\pi_{t'}} \tag{6}$$

Since we do not have access to the computed embeddings and to simplify the analysis, we assume that $\text{Var}(M_t) = 1$ from now on. One can see that $\text{Var}(H'_s)$ is minimized when $\pi_t = p, \forall t \to s$ under the constraint $\sum_{t \to s} \pi_t \le p d_s$ for some constant $p \in [0, 1]$, hence any deviation from uniformity increases the variance. The variance is also smaller the larger $\tilde{d}_s$ is. However, in theory and in practice, there is no guarantee that each vertex $s \in S$ will get any neighbors in $T$, not to mention equal numbers of neighbors. Some vertices will have pretty good estimators with thousands of samples and very low variances, while others might not even get a single neighbor sampled. For this reason, we designed LABOR so that every vertex in $S$ will sample enough neighbors in expectation.

While LADIES is optimal from an approximate matrix multiplication perspective Chen et al. (2022), it is far from optimal in the case of nonlinearities and multiple layers. Even if there is a single layer, then the used loss functions are nonlinear. Moreover, the existence of nonlinearities in-between layers and the fact that there are multiple layers exacerbates this issue and necessitates that each vertex gets a good enough estimator with low enough variance. Also, LADIES gives a formulation using sampling with replacement instead of without replacement and that is sub-optimal from the variance perspective while its implementation uses sampling without replacement without taking care of the bias created thereby. In the next section, we will show how all of these problems are addressed by our newly proposed poisson sampling framework and LABOR sampling.

## 3 LABOR: LAYER NEIGHBOR SAMPLING

As mentioned previously, node-based sampling methods suffer from sampling too shallow subgraphs leading to NEP in just a few hops (e.g., see Table 2). Layer sampling methods Zou et al. (2019) attempt to fix this by sampling a fixed number of vertices in each layer, however they can not ensure that the estimators for the vertices are of high quality, and it is hard to reason how to choose the number of vertices to sample in each layer. The original paper of LADIES Zou et al. (2019) proposes using the same number for each layer while papers evaluating it found it is better to sample an increasing number of vertices in each layer Liu et al. (2020); Chen et al. (2022). There is no systematic way to choose how many vertices to sample in each layer for the LADIES method, and since each graph has different density and connectivity structure, this choice highly depends on the graph in question. Therefore, due to its simplicity and high quality results, Neighbor Sampling currently seems to be the most popular sampling approach and there exists high quality implementations on both CPUs and GPUs in the popular GNN frameworks Wang et al. (2019); Fey & Lenssen (2019).

We propose a new approach that combines the advantages of layer and neighbor sampling approaches using a vertex-centric variance based framework, reducing the number of sampled vertices

drastically while ensuring the training quality does not suffer and matches the quality of neighbor sampling. Another advantage of our method is that the user only needs to choose the batch size and the fanout hyperparameters as in the Neighbor Sampling approach, the algorithm itself then samples the minimum number of vertices in the later layers in an unbiased way while ensuring each vertex gets enough neighbors and a good approximation.

## 3.1 LABOR SAMPLING

The design philosophy of LABOR Sampling is to create a direct alternative to Neighbor Sampling while incorporating the advantages of layer sampling. In layer sampling, the main idea can be summarized as individual vertices making correlated decisions while sampling their neighbors, because in the end if a vertex $t$ is sampled, all edges $t \to s$, into the seed vertices $S$ ($s \in S$) are added to the sampled subgraph. This can be interpreted as vertices in $S$ making a collective decision on whether to sample $t$.

The other thing to keep in mind is that, the existing layer sampling methods use sampling with replacement when doing importance sampling with unequal probabilities, because it is nontrivial to compute the inclusion probabilities in the without replacement case. The Hajek estimator in the without replacement case with equal probabilities becomes:

$$H_s'' = \frac{\sum_{t \in T \cap N(s)} \frac{M_t}{\bar{\pi}_t}}{\sum_{t \in T \cap N(s)} \frac{1}{\bar{\pi}_t}} = \frac{\sum_{t \in T \cap N(s)} M_t |N(S)|}{\sum_{t \in T \cap N(s)} |N(S)|} = \frac{1}{\tilde{d}_s} \sum_{t \in T \cap N(s)} M_t \quad (7)$$

and it has the variance:

$$\text{Var}(H_s'') = \frac{d_s - \tilde{d}_s}{d_s - 1} \frac{1}{\tilde{d}_s} \quad (8)$$

Keeping these two points in mind, we use Poisson Sampling and design LABOR sampling around it. First, let us show how one can do layer sampling using Poisson sampling (PLADIES). Given probabilities $\pi_t \in [0,1], \forall t \in N(S)$ so that $\sum_{t \in N(S)} \pi_t = n$, we include $t \in N(S)$ in our sample $T$ with probability $\pi_t$ by flipping a coin for it, i.e., we sample $r_t \sim U(0,1)$ and include $t \in T$ if $r_t \leq \pi_t$. In the end, $E[|T|] = n$ and we can still use the Hajek estimator $H_s''$ or the Horvitz Thomson estimator $H_s'$ to estimate $H_s$. This way of doing layer sampling is unbiased by construction and achieves the same goal in linear time in constrast to the quadratic time debiasing approach explained in Chen et al. (2022). In this case, the variance becomes (Williams et al., 1998):

$$\text{Var}(H_s'') \leq \text{Var}(H_s') = \frac{1}{d_s^2} \sum_{t \to s} \frac{1}{\pi_t} - \frac{1}{d_s} \quad (9)$$

One can notice the existence of the minus term $\frac{1}{d_s}$, and it enables the variance to converge to $0$ if all $\pi_t = 1$ and we get the exact result. However, in the sampling with replacement case, the variance goes to $0$ only as the sample size goes to infinity.

This type of mimicking Layer Sampling with Poisson Sampling still has the disadvantage that $\tilde{d}_s$ varies wildly for different $s$. To overcome this and mimic Neighbor Sampling where $E[\tilde{d}_s] = \min(d_s, k)$, where $k$ is a given fanout hyperparameter, we proceed as follows: for given $\pi_t \geq 0, \forall t \in N(S)$ denoting unnormalized probabilities, for a given $s$, let us define $c_s$ as the quantity satisfying the following equality if $k < d_s$, otherwise $c_s = \max_{t \to s} \frac{1}{\pi_t}$:

$$\frac{1}{d_s^2} \sum_{t \to s} \frac{1}{\min(1, c_s \pi_t)} - \frac{1}{d_s} = \frac{1}{k} - \frac{1}{d_s} \quad (10)$$

Note that $\frac{1}{k} - \frac{1}{d_s}$ is the variance when $\pi_t = \frac{k}{d_s}, \forall t \in N(s)$ so that $E[\tilde{d}_s] = k$. Also note that:

$$\frac{1}{k} - \frac{1}{d_s} - \frac{d_s - k}{d_s - 1} \frac{1}{k} = \frac{d_s - k}{k d_s} - \frac{d_s - k}{d_s - 1} \frac{1}{k} = \frac{d_s - k}{k} \left( \frac{1}{d_s} - \frac{1}{d_s - 1} \right) < 0 \quad (11)$$

meaning that the variance target we set through Eq. 10 is strictly better than Neighbor Sampling's variance in Eq. 8 and it will result in $E[\tilde{d}_s] \geq k$ with strict equality in the uniform probability case.

Then each vertex $s \in S$ samples $t \to s$ with probability $c_s \pi_t$. To keep the collective decision making, we sample $r_t \sim U(0, 1), \forall t \in N(S)$ and vertex $s$ samples vertex $t$ if and only if $r_t \leq c_s \pi_t$. Note that if we use a uniform random variable for each edge $r_{ts}$ instead of each vertex $r_t$, and if $\pi$ is uniformly initialized, then we get the same behaviour as Neighbor Sampling.

## 3.2 Importance Sampling

Given the sampling procedure above, one wonders how different choices of $\pi \geq 0$ will affect $|T|$, the total number of unique vertices sampled. In our case, it is extremely easy to compute:

$$E[|T|] = \sum_{t \in N(S)} \mathbb{P}(t \in T) = \sum_{t \in N(S)} \min(1, \pi_t \max_{t \to s} c_s) \tag{12}$$

In particular, we need to find $\pi^* \geq 0$ minimizing $E[|T|]$:

$$\pi^* = \arg\min_{\pi \geq 0} \sum_{t \in N(S)} \min(1, \pi_t \max_{t \to s} c_s) \tag{13}$$

Note that for any given $\pi \geq 0$, $E[|T|]$ is the same for any vector multiple $x\pi, x \in \mathbb{R}^+$, meaning that the objective function is homogenous of degree 0.

## 3.3 Computing $c$ and $\pi^*$

Note that $c_s$ was defined to be the scalar satisfying the following equality involving the variance of the estimator of $H_s$:

$$\frac{1}{d_s^2} \sum_{t \to s} \frac{1}{\min(1, c_s \pi_t)} - \frac{1}{d_s} = \frac{1}{k} - \frac{1}{d_s} \tag{14}$$

If we rearrange the terms, we get:

$$\sum_{t \to s} \frac{1}{\min(1, c_s \pi_t)} = \frac{d_s^2}{k} \tag{15}$$

One can see that the left hand side of the equality is monotonically decreasing with respect to $c_s \geq 0$. Thus one can use binary search to find the $c_s$ satisfying the above equality to any precision needed. But we opt to use the following iterative algorithm to compute it:

$$v_s^{(0)} = 0, c_s^{(0)} = \frac{k}{d_s^2} \sum_{t \to s} \frac{1}{\pi_t} \tag{16}$$

$$c_s^{(i+1)} = \frac{c_s^{(i)}}{\frac{d_s^2}{k} - v_s^{(i)}} \left( -v_s^{(i)} + \sum_{t \to s} \frac{1}{\min(1, c_s^{(i)} \pi_t)} \right), v_s^{(i+1)} = \sum_{t \to s} \mathbb{1}[c_s^{(i+1)} \pi_t \geq 1] \tag{17}$$

This iterative algorithm converges in at most $d_s$ steps and the convergence is exact and monotonic from below. One can also implement it in linear time $\mathcal{O}(d_s)$ if $\{\pi_t \mid t \to s\}$ is sorted and making use of precomputed prefix sum arrays. Note that $c = c(\pi)$, meaning that $c$ is a function of the given probability vector $\pi$. To compute $\pi^*$, we use a similar fixed point iteration as follows:

$$\pi^{(0)} = 1, \forall t \in N(S) : \pi_t^{(i+1)} = \pi_t^{(i)} \max_{t \to s} c_s(\pi^{(i)}) \tag{18}$$

Thus, we alternate between computing $c = c(\pi)$, meaning $c$ is computed with the current $\pi$, and updating the $\pi$ with the computed $c$ values. Each step of this iteration is guaranteed to lower the objective function value in Eq. 13 until convergence to a fixed point, see the Appendix A.1. Modified formulation for a given nonuniform weight matrix $A_{ts}$ is explained in the Appendix A.3.

Table 1: Datasets used in experiments, numbers of vertices, edges, avg. degree, features, sampling budget used, training, validation and test vertex split.

| Dataset | $|V|$ | $|E|$ | $\frac{|E|}{|V|}$ | # feats. | $|V_3|$ budget | train - val - test (%) |
|---------|-------|-------|-------------------|----------|----------------|------------------------|
| reddit | 233K | 115M | 493.56 | 602 | 60k | 66 - 10 - 24 |
| products | 2.45M | 61.9M | 25.26 | 100 | 400k | 8 - 2 - 90 |
| yelp | 717K | 14.0M | 19.52 | 300 | 200k | 75 - 10 - 15 |
| flickr | 89.2K | 900K | 10.09 | 500 | 70k | 50 - 25 - 25 |

## 3.4 CHOOSING HOW MANY NEIGHBORS TO SAMPLE

The variance of Poisson Sampling when $\pi_t = \frac{k}{d_s}$ is $\frac{1}{k} - \frac{1}{d_s}$. One might question why we are trying to match the variance of Neighbor Sampling and choose to use a fixed fanout for all the seed vertices. In the uniform probability case, if we have already sampled some set of edges for all vertices in $S$, and want to sample one more edge, the question becomes which vertex in $S$ should we sample the new edge for? Our answer to this question is the vertex $s$, whose variance would improve the most. If currently vertex $s$ has $\tilde{d}_s$ edges sampled, then sampling one more edge for it would improve its variance from $\frac{1}{\tilde{d}_s} - \frac{1}{d_s}$ to $\frac{1}{1+\tilde{d}_s} - \frac{1}{d_s}$. Since the derivative of the variance with respect to $\tilde{d}_s$ is monotonic, we are allowed to reason about the marginal improvements by comparing their derivatives, which is:

$$\frac{\partial(\frac{1}{\tilde{d}_s} - \frac{1}{d_s})}{\partial \tilde{d}_s} = -\frac{1}{\tilde{d}_s^2} \tag{19}$$

Notice that the derivative does not depend on the degree $d_s$ of the vertex $s$ at all, and the greater the magnitude of the derivative, the more improvement the variance of a vertex gets by sampling one more edge. Thus, choosing any vertex $s$ with least number of edges sampled would work for us, that is: $s = \arg\min_{s' \in S} \tilde{d}_{s'}$. In light of this observation, one can see that it is optimal to sample an equal number of edges for each vertex in $S$. This is one of the reasons LADIES is not efficient with respect to the number of edges it samples. On graphs with skewed degree distributions, it samples thousands of edges for some seed vertices, which contribute very small amounts to the variance of the estimator since it is already very close to $0$.

## 4 EXPERIMENTS

In this section, we empirically evaluate the performance of each method in the node-prediction setting on the following datasets: reddit (Hamilton et al., 2017), products (Hu et al., 2020a), yelp, flickr (Zeng et al., 2020). Details about these datasets are given in Table 1. We compare LABOR variants LABOR-0, LABOR-1 and LABOR-*, where $0, 1, *$ stand for the number of fixed point iterations applied to optimize 13 respectively, against NS (Neighbor Sampling), LADIES and PLADIES sampling methods, where PLADIES is the unbiased Poisson Sampling variant of LADIES introduced in Section 3.1. We do not include Fast-GCN in our comparisons as it is superseeded by the LADIES paper. The works of Liu et al. (2020); Zhang et al. (2021); Huang et al. (2018); Cong et al. (2021); Dong et al. (2021) are not included in the comparisons because they either take into account additional information such as historical embeddings or their magnitudes or they have a different sampling structure such as a vertex cache to sample from. Also the techniques in these papers are mostly orthogonal to the sampling problem and algorithms discussed in this paper. We evaluate all the baselines on the GCN model in Eq. 2 with 3 layers, with 256 hidden dimension and residual skip connections enabled. We use the Adam optimizer (Kingma & Ba, 2014) with $0.001$ learning rate. We carried out our experiments using the DGL framework (Wang et al., 2019) with the Pytorch backend (Paszke et al., 2019) [1]. Experiments were repeated 100 times and averages are presented.

We will first show that despite the different number of sampled vertices, LABOR and NS training loss curves are almost the same in Section 4.1 with the same fanout and batch size hyperparameters. We will match the hyperparameters of LADIES with the number of vertices sampled on LABOR and

---

[1]The code will be contributed to the DGL framework after the blind review process.

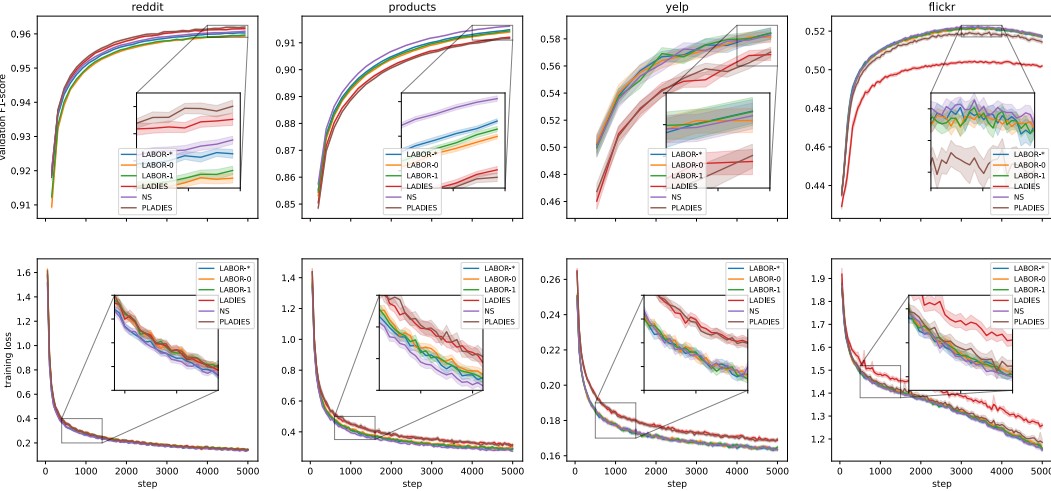

Figure 1: The validation F1-score and training loss curves on different datasets with same batch size. The soft edges represent the confidence interval. Number of sampled vertices and edges can be found in Table 2.

see whose batches have better quality. Then, we will show what happens when different sampling algorithms are given the same budget and compare their vertex sampling efficiency in Section 4.2. Section 4.3 shows the reduction in the number of vertices sampled with each fixed point iteration.

## 4.1 COMPARISON AGAINST NEIGHBOR SAMPLING AND LADIES

In this experiment, we set the batch size to 1,000 and the fanout $k = 10$ for LABOR and NS methods to see the difference in the sizes of the sampled subgraphs and also whether convergence behaviour is the same. In Figure 1, we can see that the convergence curves of both NS and LABOR variants are almost the same showing that sampling smaller subgraphs does not really affect the batch quality. Table 2 shows the difference of the sampled subgraph sizes in each layer. One can see that on reddit, LABOR-* samples $6.9\times$ fewer vertices in the 3rd layer while keeping the same convergence behaviour. On the flickr dataset however, LABOR-* samples only $1.3\times$ fewer vertices. The amount of difference depends on two factors. The first is the amount of overlap of neighbors among the vertices in $S$. If the neighbors of vertices in $S$ did not overlap at all, then one obviously can not do better than NS. The second is the average degree of the graph. With a fanout of $10$, both Neighbor Sampling and LABOR has to copy the whole neighborhood of a vertex $s$ with degree $d_s \leq 10$. Thus for such graphs, it is expected that there is a small difference because for a lot of the vertices, their whole neighborhood is copied. If we look at Table 1, the average degree of the flickr graph is $10.09$, and thus there is only a small difference between LABOR and NS.

In Table 2, the number of sampled edges is another important metric to look at. We can see that LABOR-0 reduces both the number of vertices and edges sampled. On the other hand, when importance sampling is enabled, the number of vertices sampled goes down while number of edges sampled goes up. This is because when importance sampling is used, inclusion probabilities become nonuniform and it takes more edges per seed vertex to get a good approximation (see Eq. 10).

The hyperparameters of LADIES and PLADIES were picked to match LABOR-* so that all methods have the same sampling budget in each layer (see Table 2). Figure 1 shows that, in terms of loss curve, LADIES and PLADIES perform almost the same on all but the flickr dataset, in which case there is a big difference between the two in favor of PLADIES. We also see that LABOR variants either match the quality of PLADIES on reddit or outperform it on products, yelp and flickr.

Looking at Table 2, we can see that LABOR-0 has the best runtime performance across all datasets. This is both due to lack of the overhead of performing the fixed point iterations and also it samples the fewest edges, compared to the other LABOR variants. By design, all LABOR variants should have the same convergence curves, as seen in Figure 1. Then, the decision of which variant to use depends on one factor: feature access speed. If vertex features were stored on a slow storage medium

Table 2: Average number of vertices and edges sampled in different layers (All the numbers are in thousands, lower is better). Last column shows iterations(minibatches) per second (it/s) (higher is better). The hyperparameters of LADIES and PLADIES were picked to roughly match the number of vertices sampled by the LABOR-* to get a fair comparison. The convergence curves can be found in Figure 1. The timing information was measured on an NVIDIA T4 GPU. Green stands for best, red stands for worst results, with a 5% cutoff.

| Dataset | Algo. | $|V_3|$ | $|E_3|$ | $|V_2|$ | $|E_2|$ | $|V_1|$ | $|E_1|$ | $|V_0|$ | it/s |
|---|---|---|---|---|---|---|---|---|---|
| reddit | PLADIES | **24.0** | **2390** | 14.1 | 927 | 5.97 | 33.2 | 1 | **1.7** |
| | LADIES | **25.3** | **2270** | 14.5 | 852 | 5.97 | 32.5 | 1 | **1.8** |
| | LABOR-* | **24.1** | 1070 | 13.7 | 435 | 6.03 | 26.9 | 1 | 4.1 |
| | LABOR-1 | 26.6 | 261 | 14.4 | 116 | 6.12 | 16.7 | 1 | 24.8 |
| | LABOR-0 | 35.8 | **177** | 17.8 | 67.1 | 6.77 | 9.64 | 1 | **37.6** |
| | NS | **167** | 682 | 68.3 | 100 | 10.1 | 9.65 | 1 | 14.2 |
| products | PLADIES | **160** | **2380** | 51.2 | 293 | 9.68 | 11.7 | 1 | **4.1** |
| | LADIES | **165** | **2230** | 51.8 | 270 | 9.68 | 11.5 | 1 | **4.2** |
| | LABOR-* | **166** | 1250 | 51.8 | 167 | 9.78 | 10.6 | 1 | 6.2 |
| | LABOR-1 | 178 | 799 | 53.4 | 136 | 9.78 | 10.5 | 1 | 21.3 |
| | LABOR-0 | 237 | **615** | 62.4 | 99.6 | 10.1 | 9.89 | 1 | **32.5** |
| | NS | **513** | 944 | 95.4 | 106 | 10.6 | 9.89 | 1 | 24.6 |
| yelp | PLADIES | **100** | **1300** | 29.5 | 183 | 6.16 | 6.90 | 1 | **5.1** |
| | LADIES | **102** | **1280** | 29.7 | 182 | 6.17 | 6.90 | 1 | **5.3** |
| | LABOR-* | **105** | 991 | 30.7 | 158 | 6.15 | 6.83 | 1 | 13.3 |
| | LABOR-1 | 109 | 447 | 31.0 | 96.1 | 6.18 | 6.76 | 1 | **27.3** |
| | LABOR-0 | 138 | **318** | 35.1 | 53.9 | 6.25 | 6.29 | 1 | **27.2** |
| | NS | **188** | 392 | 42.5 | 54.8 | 6.32 | 6.27 | 1 | 23.0 |
| flickr | PLADIES | **55.2** | **309** | 24.9 | 84.7 | 6.23 | 6.89 | 1 | **10.2** |
| | LADIES | **55.9** | **308** | 25.1 | 84.5 | 6.23 | 6.89 | 1 | **10.5** |
| | LABOR-* | **56.6** | **308** | 25.6 | 84.7 | 6.29 | 6.95 | 1 | 20.3 |
| | LABOR-1 | 57.7 | 242 | 25.9 | 73.4 | 6.29 | 6.93 | 1 | **32.7** |
| | LABOR-0 | 65.9 | **219** | 29.1 | 51.9 | 6.37 | 6.71 | 1 | **33.3** |
| | NS | **73.3** | 244 | 32.8 | 51.9 | 6.37 | 6.72 | 1 | **31.7** |

(such as, on host memory accessed over PCI-E), then minimizing number of sampled vertices would become the highest priority, in which case, one should pick LABOR-*. Depending on the relative vertex feature access performance and the performance of the training processor, one can choose to use LABOR-$j$, the faster feature access, the lower the $j$.

## 4.2 EVALUATION OF VERTEX SAMPLING EFFICIENCY

In this experiment, we set a limit on the number of sampled vertices and modify the batch size to match the given vertex budget. The budgets used were picked around the same magnitude with numbers in the Table 2 in the $|V_3|$ column and can be found in Table 1. Figure 2 displays the result of this experiment. Table 3 shows that the more vertex efficient the sampling method is, the larger batch size it can use during training. Number of sampled vertices is not a function of the batch size for the LADIES algorithm so we do not include it in this comparison. All of the experiments were repeated 100 times and their averages were plotted, that is why our convergence plots are smooth and differences are clear. The most striking result in this experiment is that there can be upto $112\times$ difference in batch sizes of LABOR-* and NS algorithms on the reddit dataset, which translates into faster convergence as the training loss and validation F1-score curves in Figure 2 show.

## 4.3 IMPORTANCE SAMPLING, NUMBER OF FIXED POINT ITERATIONS

In this section, we look at the convergence behaviour of the fixed point iterations described in Section 3.3. Table 4 shows the number of sampled vertices in the last layer with respect to the number of fixed point iterations applied. In this table, the $\infty$ stands for applying the fixed point iterations

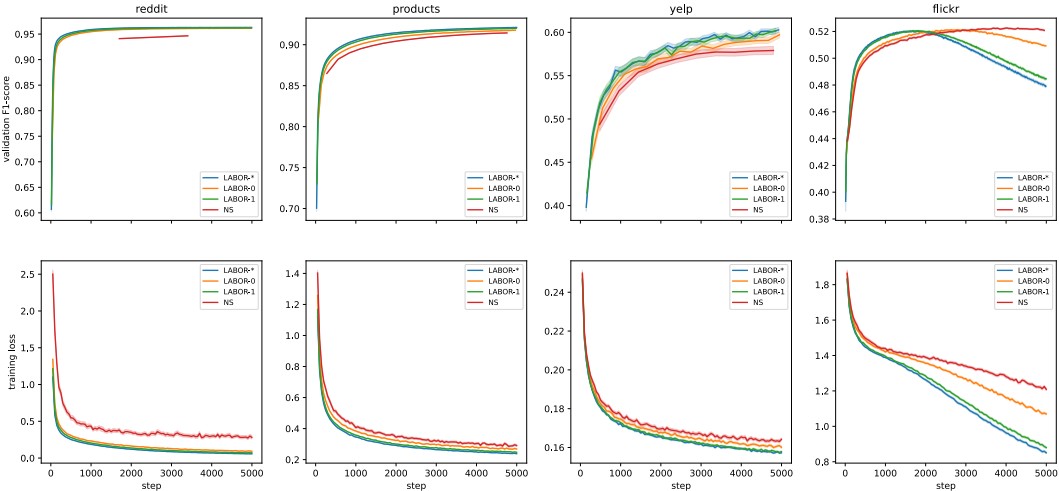

Figure 2: Validation F1-score and the training loss curves to evaluate vertex sampling efficiency under the same sampling budget. The batch size is chosen so that the # sampled vertices matches the vertex budget for each dataset and method and it is given in Table 3.

Table 3: The batch sizes used in Figure 2. These were chosen such that in expectation, each method samples with the same budget given in Table 1. Having a larger batch-size speeds up convergence.

| Dataset | LABOR-* | LABOR-1 | LABOR-0 | NS |
|---|---|---|---|---|
| reddit | **10100** | 8250 | 3850 | **90** |
| products | **7750** | 6000 | 2750 | **700** |
| yelp | **3700** | 3300 | 1950 | **1120** |
| flickr | **3600** | 3100 | 1400 | **800** |

until convergence, and convergence occurs in at most 15 iterations in practice before the relative change in the objective function is less than $10^{-4}$. One can see that most of the reduction in the objective function 13 occurs after the first iteration, and the remaining iterations have diminishing returns. Full convergence can save from $14\%$ - $33\%$ depending on the dataset. The monotonically decreasing numbers provide empirical evidence for the presented proof in the Appendix A.1.

Table 4: Number of vertices (in thousands) in 3rd layer w.r.t # fixed point iterations (its). $\infty$ denotes applying the fixed point iterations until convergence, i.e., LABOR-*, 1 its stands for LABOR-1 etc.

| Dataset | NS | 0 its | 1 its | 2 its | 3 its | $\infty$ its |
|---|---|---|---|---|---|---|
| reddit | 167 | 35.8 | 26.5 | 25.0 | 24.7 | 24.2 |
| products | 513 | 237 | 178 | 170 | 169 | 166 |
| yelp | 188 | 138 | 109 | 106 | 105 | 105 |
| flickr | 73.3 | 65.9 | 57.6 | 56.8 | 56.6 | 56.5 |

## 5 CONCLUSIONS

In this paper, we introduced LABOR sampling, a novel way to combine layer and neighbor sampling approaches using a vertex-variance centric framework. We then transform the sampling problem into an optimization problem where the constraint is to match neighbor sampling variance for each vertex while sampling the fewest number of vertices. We show how to minimize this new objective function via fixed-point iterations. On datasets with dense graphs like Reddit, we show that our approach can sample a subgraph with $7\times$ fewer vertices without degrading the batch quality. We also show that compared to LADIES, LABOR converges faster with same sampling budget.

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

# A  APPENDIX

## A.1  FIXED POINT ITERATIONS

Given any $\pi^{(0)} > 0$ and one iteration to get $\pi^{(1)}$ and one more iteration using $c(\pi^{(1)})$ to get $\pi^{(2)}$, then we have the following observation:

$$\pi_t^{(1)} = \pi_t^{(0)} \max_{t \to s} c_s(\pi^{(0)})$$

$$\frac{1}{d_s^2} \sum_{t \to s} \frac{1}{\min(1, c_s(\pi^{(0)})\pi_t^{(0)})} - \frac{1}{d_s} = \frac{1}{k} - \frac{1}{d_s}$$

$$\frac{1}{d_s^2} \sum_{t \to s} \frac{1}{\min(1, c_s(\pi^{(1)})\pi_t^{(1)})} = \frac{1}{d_s^2} \sum_{t \to s} \frac{1}{\min(1, c_s(\pi^{(1)}) \max_{t \to s'} c_{s'}(\pi^{(0)})\pi_t^{(0)})}$$

Now, note that for a given $t \in N(s)$, $\max_{t \to s'} c_{s'}(\pi^{(0)}) \geq c_s(\pi^{(0)})$ since $s \in \{s' \mid t \to s'\}$. This implies that $\max_{t \to s'} c_{s'}(\pi^{(0)})\pi_t^{(0)} \geq c_s(\pi^{(0)})\pi_t^{(0)}$. Note that for $\pi_t' = c_s(\pi^{(0)})\pi_t^{(0)}, \forall t \to s$, we have $c_s(\pi') = 1$ for any given $s$. Since $\max_{t \to s'} c_{s'}(\pi^{(0)})\pi_t^{(0)} \geq c_s(\pi^{(0)})\pi_t^{(0)} = \pi_t', \forall t \to s$, this let's us conclude that the $c_s(\pi^{(1)}) \leq 1$, because the expression is monotonically increasing with respect to any of the $\pi_t$. By induction, this means that $c_s^{(i)} \leq 1, \forall i \geq 1$. Since $\pi_t^{(i)} = \pi_t^{(0)} \prod_{j=0}^{i-1} \max_{t \to s'} c_{s'}(\pi^{(i-1)})$, $\pi_t^{(i)}$ is monotonically decreasing. This means that the objective value in Eq. 13 is also monotonically decreasing and is clearly bounded from below by 0. Any monotonically decreasing sequence bounded from below has to converge, so our fixed point iteration procedure is convergent as well.

The intuition behind the proof above is that after updating $\pi$ via Eqn. 18, the probability of each edge $t \to s : \pi_t c_s$ goes up because the update takes the maximum $c_s$ over all possible $s$. This means each vertex gets higher quality estimators than the set variance target so we now have room to reduce the number of vertices sampled by choosing the appropriate $c_s \leq 1$. To summarize, $\pi$ update step increases the quality of the batch which in turn let's the $c_s$ step to reduce the total number of vertices sampled.

## A.2  EXACTLY SAMPLING A FIXED NUMBER OF NEIGHBORS

One can easily resort to Sequential Poisson Sampling by Ohlsson (1998) if one wants $\tilde{d}_s = \min(k, d_s)$ instead of $E[\tilde{d}_s] = \min(k, d_s)$ to get the exact same behaviour to Neighbor Sampling. Given $\pi_t$, $c_s$ and $r_t$, we pick the $\tilde{d}_s = \min(k, d_s)$ smallest vertices $t \to s$ with respect to $\frac{r_t}{c_s \pi_t}$, which can be computed in expected linear time by using the quickselect algorithm (Hoare, 1961).

## A.3  EXTENSION TO THE WEIGHTED CASE

If the given adjacency matrix $A$ has nonuniform weights, then we want the estimate the following:

$$H_s = \frac{1}{A_{*s}} \sum_{t \to s} A_{ts} M_t \tag{20}$$

where $A_{*s} = \sum_{t \to s} A_{ts}$. If we have a probability over each edge $\pi_{ts}, \forall (t \to s) \in E$, then the variance becomes:

$$\mathrm{Var}(H_s'') \leq \mathrm{Var}(nH_s') = \frac{1}{(A_{*s})^2} \left( \sum_{t \to s} \frac{A_{ts}^2}{\min(1, c_s \pi_{ts})} - \sum_{t \to s} A_{ts}^2 \right) \tag{21}$$

In this case, we can still aim to reach the same variance target $v_s = \frac{1}{k} - \frac{1}{d_s}$ or any given custom target $v_s \in \mathbb{R}^+$ by finding $c_s$ that satisfies the following equality:

$$\frac{1}{(A_{*s})^2}\Big(\sum_{t \to s} \frac{A_{ts}^2}{\min(1, c_s\pi_{ts})} - \sum_{t \to s} A_{ts}^2\Big) = v_s \tag{22}$$

In this case, the objective function becomes:

$$\pi^* = \arg\min_{\pi \geq 0} \sum_{t \in N(S)} \min(1, \max_{t \to s} c_s\pi_{ts}) \tag{23}$$

Optimizing the objective function above will result into minimizing the number of vertices sampled. Given any $\pi_{ts} > 0, \forall(t \to s)$, then the fixed point iterations proposed for the non-weighted case in Eq. 18 can be modified as follows:

$$\pi^{(0)} = A, \forall(t \to s) : \pi_{ts}^{(i+1)} = \max_{t \to s'} c_{s'}(\pi^{(i)})\pi_{ts'}^{(i)} \tag{24}$$

A more principled way to choose $v_s$ in the weighted case is by following the argument presented in Section 3.4. There, the discussion revolves around the derivative of the variance with respect to the expected number of vertices sampled for a given seed vertex $s$. If we apply the same argument, then we get:

$$v_s(c_s) = \frac{1}{(A_{*s})^2}\Big(\sum_{t \to s} \frac{A_{ts}^2}{\min(1, c_s\pi_{ts})} - \sum_{t \to s} A_{ts}^2\Big)$$

$$\frac{\partial v_s}{\partial c_s} = \frac{-1}{(A_{*s})^2}\sum_{t \to s} \frac{\mathbb{1}[c_s\pi_{ts} < 1]A_{ts}^2}{c_s^2\pi_{ts}}$$

$$E[\tilde{d}_s](c_s) = \sum_{t \to s} \min(1, c_s\pi_{ts})$$

$$\frac{\partial E[\tilde{d}_s]}{c_s} = \sum_{t \to s} \mathbb{1}[c_s\pi_{ts} < 1]\pi_{ts}$$

Then to compute the derivative of the variance with respect to the expected number of vertices sampled for a given seed vertex $s$, which is $\frac{\partial v_s}{\partial E[\tilde{d}_s]}$, we can use the chain rule and get:

$$\frac{\partial v_s}{\partial E[\tilde{d}_s]} = \frac{\partial v_s}{\partial c_s}\frac{\partial c_s}{\partial E[\tilde{d}_s]} = \frac{\frac{-1}{(A_{*s})^2}\sum_{t \to s} \frac{\mathbb{1}[c_s\pi_{ts}<1]A_{ts}^2}{c_s^2\pi_{ts}}}{\sum_{t \to s} \mathbb{1}[c_s\pi_{ts} < 1]\pi_{ts}} \tag{25}$$

Then, what one would do is to choose a constant $C(k)$ as a function of the fanout parameter $k$ and set $\frac{\partial v_s}{\partial E[\tilde{d}_s]} = C(k)$ and solve for $c_s$. $C(k)$ would be a negative quantity whose absolute value decreases as $k$ increases. It would probably look like $C(k) = \frac{-C'}{k^2}$ for some constant $C' > 0$. We leave this as future work.

### A.4 CONVERGENCE SPEED WITH RESPECT TO WALL TIME

In this section, we perform hyperparameter optimization on Neighbor Sampler and Labor Sampler so that the training converges to a target validation accuracy as fast as possible. We leave LADIES out of this experiment because it is too slow as can be seen in the last column of Table 2. We ran this experiment on an A100 GPU and stored the input features on the main memory, which were accessed over the PCI-e directly during training by pinning their memory. This kind of training scenario is commonly used when training on large datasets whose input features don't fit in the GPU memory. We use larger of the two datasets we have for this experiment, products and yelp. For

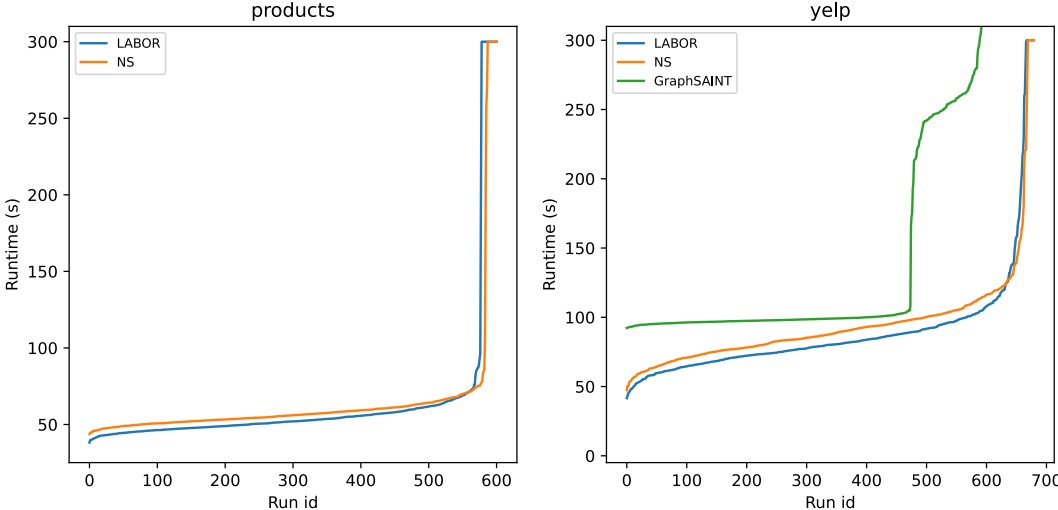

Figure 3: The runtimes to reach a validation F1-score of 91.5% on products and 60% on yelp, belonging to runs tried by the HEBO hyperparameter tuner, sorted with respect to their runtimes. Having a lower curve means that a method is faster overall compared to the others. HEBO could not find any hyperparameter configuration to reach the set target on products, hence its curve is left out.

products, the validation accuracy target we set is 91.5%, and for yelp it is 60%. We tune the learning rate between $[10^{-4}, 10^{-1}]$, the batch size between $[2^{10}, 2^{15}]$ and the fanout for each layer between $[5, 25]$. We use the same model used in Section 4. For LABOR, we additionally tune over the number of importance sampling iterations i between $[0, 3]$ so that it can switch between LABOR-$i$ and also a layer dependency boolean parameter that makes LABOR use the same random variates $r_t$ for different layers when enabled, which has the effect of increasing the overlap of sampled vertices across layers. We use the state of the art hyperparameter tuner HEBO Cowen-Rivers et al. (2020), the winning submission to the NeurIPS 2020 Black-Box Optimisation Challenge, to tune the parameters of the sampling algorithm with respect to runtime required to reach the target validation accuracy with a timeout of 300 seconds, terminating the run if the configuration doesn't reach the target accuracy.

We let HEBO run overnight and collect the minimum runtimes required to achieve the target accuracies. For products, the fastest hyperparameter corresponding to the 38.2s runtime had fanouts $(18, 5, 25)$, batch size $10500$, learning rate $0.0145$, used LABOR-1, layer dependency False.

For Neighbor Sampler, the fastest hyperparameter corresponding to 43.82s runtime had fanouts $(15, 5, 21)$, batch size $12000$, learning rate $0.0144$.

For the Yelp dataset, the fastest hyperparameter corresponding to the 41.60s runtime had fanouts $(6, 5, 7)$, batch size $5400$, learning rate $0.000748$, used LABOR-1 and layer dependency True.

For Neighbor Sampler, the fastest hyperparameter corresponding to the 47.40s runtime had fanouts $(5, 6, 6)$, batch size $4600$, learning rate $0.000931$.

These results clearly indicate that training with LABOR is faster compared to Neighbor Sampling when it comes to time to convergence.

We run the same experiment with GraphSAINT (Zeng et al., 2020) using the DGL example code, both on ogbn-products and yelp using the same model architecture. We used their edge sampler, the edge sampling budget between $[2^{10}, 2^{15}]$ and learning rate between $[10^{-4}, 10^{-1}]$. We disabled batch normalization to make it a fair comparison with NS and LABOR since the models they are using does not have batch normalization. We use HEBO to tune these hyperparameters to reach the set validation accuracy, and let it run overnight. The results show that HEBO was not able to find any configuration of the hyperparameters reaching 91.5% accuracy on products faster than 1500 seconds. For the Yelp dataset, the fastest runtime to reach 60% accuracy was 92.2s, with an edge sampling budget of $12500$ and learning rate $0.0214$.

