# OpenReview forum: "(LA)YER-NEIGH(BOR) SAMPLING: DEFUSING NEIGHBORHOOD EXPLOSION"
_ICLR.cc/2023/Conference — Submitted to ICLR 2023_

### Official Review · Reviewer_M7QF · 2022-10-23

**Confidence:** 3
**Correctness:** 4
**Technical Novelty And Significance:** 3
**Empirical Novelty And Significance:** 2
**Recommendation:** 5

**Clarity, Quality, Novelty And Reproducibility:**

The pipeline is well explained with a solid explanation of the theoretical foundation. However, the experimental results are not conclusive enough to state the usefulness of the method.

**Strength And Weaknesses:**

Strengths:
1. This work addresses the Neighborhood Explosion Phenomenon (NEP), the curse of dimensionality version of Graph Network Training.
Weaknesses:
1. The abstract states the method's superiority but not by how much or against what the method is compared.
2. The same happens in the main body of the work: it is not clearly stated how much better is the proposed pipeline.
3. The last point is important because the proposed method depends on a long series of assumptions and results that are true in general or valid in limit cases. Thus, the actual effectiveness of the method, which could be shown with experiments, is not indeed shown. Thus, it is challenging to state whether the method is suitable.

**Summary Of The Paper:**

The authors present LABOR, a method that experimentally shows to be better than LABOR neighbor sampling and layer sampling approaches.

**Summary Of The Review:**

It is an interesting paper with a great deal of theoretical justification. However, the fundaments rest on general assumptions whose usefulness needs to be assessed with experiments. But, as presented, these experiments are not conclusive enough to judge the pipeline benefits.

---

> ### Author Response · Authors · 2022-11-18
> **Summarizing benefits of using LABOR compared to NS and LADIES, improvements to the main text.**
>
> Thank you very much for your feedback.
>
> * We have updated the abstract with numbers showing how much improvement we get compared to NS, that is LABOR can sample upto 7x less compared to NS and can use 112 times larger batch-size compared to NS. With the same vertex sampling budget, it also converges more than 3 times faster than LADIES variants (see Table 2).
>
> * We have also updated the contributions paragraph in main text at the end of the introduction with concrete numbers such as 2.6 times faster convergence.
>
> * To summarize the benefits of our method, since it is a drop-in replacement to NS, it can sample upto 7x fewer vertices, speed up convergence upto 2.6x, and can use a 112x larger batch size compared to NS, (see Table 2 and Table 3). We are hoping that these improvements are enough to show that LABOR is effective and suitable and can be used as a drop-in replacement for NS as Figures 1 and 2 show. Also, Table 2 shows that LABOR needs upto 13.5x fewer edges to converge compared to LADIES.

---

### Official Review · Reviewer_5q4F · 2022-10-24

**Confidence:** 5
**Correctness:** 3
**Technical Novelty And Significance:** 2
**Empirical Novelty And Significance:** 2
**Recommendation:** 3

**Clarity, Quality, Novelty And Reproducibility:**

Overall, the paper is clear and written with a high quality. My main concerns are the novelty and the experimental justification.  I have no idea of the reproducibility since there is no source code released, but I think the experiments are easy to reproduce.

**Details Of Ethics Concerns:**

None.

**Strength And Weaknesses:**

## Strengths
 - Well organized and easy to follow.
 - Detailed analyze about the sampling probability with   fanout hyperparameter $k$.

## Weakness
 - Technical contribution is limited. Principally, This paper follow the layer-wise sampling paradigm and replace the uniform sampling with the Poisson sampling. The paper also show that this sampling strategy can obtain a smaller sampling variance. This point seems a marginal improvement over the existing sampling framework. Concretely, except for the smaller sampling variance, I can not figure out the benefits of adopting this sampling strategy in the large-scale GNN training from the analysis in Section 3. From the theoretical perspective, would it improve the convergence speed  or reduce the memory cost, or other benefits?

 Since the efficient training
 - Lack of comparison with other layer-wise baselines. As mentioned in Introduction, there are several layer-wise sampling algorithms, such as AS-GCN[Huang et al., 2018] and GNS[Dong et al., 2021] (Both of them have released the source code.). However, this paper only make the comparison with LADIES. This is not sufficient  considering both studies have released source codes.

 - The experiments can not demonstrate the superiority of the proposed method. First, the paper only shows the training loss curve of all the methods. This is not sufficient since we can not get any cue about how the sampling strategy would affect the generalization ability unless the validation curves or the metrics on the test set are reported. Despite that, the improvement over the training loss is still very marginal and we can only observe its difference according to the zoom-in figure.

 - Moreover, to demonstrate the effectiveness of the proposed method. It's better to make such comparisons:
   - Vanilla GNN model with full-batch on the small data: to check the performance gain/drop of the proposed sampling method.
   - Other kinds of sampling strategies, e.g. GraphSAINT [1] and ClusterGCN [2].



 [1] Zeng, Hanqing, et al. "Graphsaint: Graph sampling based inductive learning method." arXiv preprint arXiv:1907.04931 (2019).

 [2] Chiang, Wei-Lin, et al. "Cluster-gcn: An efficient algorithm for training deep and large graph convolutional networks." Proceedings of the 25th ACM SIGKDD international conference on knowledge discovery & data mining. 2019.

**Summary Of The Paper:**

This paper proposed a new sampling  strategy: LABOR, which employs the Poisson Sampling to achieve a better sampling variance. Furthermore, it introduces a fanout hyperparameter $k$ and  delicate importance sampling probability to sample much fewer vertices without sacrificing quality.

**Summary Of The Review:**

This paper is about a new sampling strategy of the layer-wise sampling in the GNN training. The details are clear and the analysis of sampling parameters is OK. However, the contribution of this  modification  looks marginal and the experiments failed to demonstrate the superiority of the proposed method. I vote the rejection for this paper.

---

> ### Author Response · Authors · 2022-11-18
> **Added new experimental results, explanation of experimental setup.**
>
> Thank you very much for your feedback. We would like to point out that NS has very optimized CPU and GPU implementations in the DGL framework and other frameworks and is the defacto sampling algorithm to use. Our results in Table 2 show that on an NVIDIA T4 GPU, using our method with the same batch size and fanout hyperparameters, training can speed up upto 2.6 times. Since the convergence behaviour is similar to NS, this is a free speedup gained just by replacing NS with LABOR-0 along with upto 7x less memory use per batch. Our general comment explains why comparisons against AS-GCN and GNS are lacking, because the approaches they are proposing are orthogonal and can be applied to LABOR also.
>
> We have also included the validation F1-score curves and they show the same trend as the training loss curves. Figure 2 validation accuracy plots show that our method is by far ahead in the convergence compared to NS. Being able to use a larger batch size under vertex sampling budget constraints, our method converges more than 2 times faster compared to NS on almost all datasets and there is upto 3% difference in accuracy.
>
> We haven't yet included an experiment comparing against full-batch training but will do so for the camera ready version. LABOR will enjoy the same benefits NS enjoys when it comes to generalization ability since they behave almost exactly the same as our experiments show.
>
> We have included an experiment the Appendix comparing our method against GraphSAINT. It shows that LABOR is faster to converge to a given target validation accuracy than NS and GraphSAINT.

---

### Official Review · Reviewer_3iA9 · 2022-10-31

**Confidence:** 3
**Correctness:** 3
**Technical Novelty And Significance:** 2
**Empirical Novelty And Significance:** 2
**Recommendation:** 3

**Clarity, Quality, Novelty And Reproducibility:**

The paper is with good writing. The sampling strategy seems novel to me.


**Strength And Weaknesses:**

Strength:
The paper is about how to perform sampling GNN training. The sampling method proposed seems a reasonable way to combine the layer and neighbor sampling methods.

Weakness:

1: The proposed method aims to replace NS using a variance reduction sampling and achieves better convergence. The question is whether NS achieves SOTA performance on the standard GNNs datasets?

2: My main concern is on the experiment part:

1) The paper is about sampling method to speed up GNN training. However, there are no result on wall time for training GNNs.

2) There is no accuracy reported for all the four datasets tested in the experiment part.

3) No comparison with latest GNN fast training methods.

4) missing ablation studies.

**Summary Of The Paper:**

This paper is about speeding up the GNN training with a new way for sampling which combines layer and neighbor sampling methods.

**Summary Of The Review:**

I think the paper is interesting.  Due to limited experiment results, I am not fully convinced by this method for speeding up GNNs training.

---

> ### Author Response · Authors · 2022-11-18
> **Explaining significance of NS, our contributions and adding additional experiments and plots**
>
> Thank you very much for your feedback. We see your point about whether NS being the SOTA method. However not only did we improve NS, but also we show that it outperforms other existing Layer sampling methods. We also propose Poisson sampling so that other Layer sampling methods can benefit from a lower variance. We show that our proposed PLADIES variant outperforms the original LADIES. Our general comment also points out that NS is commonly used in recent SOTA works. Majority of SOTA sampling methods use NS in some way. Our goal is to provide a drop-in replacement to NS with the same hyperparameters so that everyone can easily use LABOR as a replacement for NS. NS is also is the default sampling algorithm in a lot of the GNN deep learning frameworks such as DGL and Pytorch Geometric and we are  hoping that LABOR can become the new default, LABOR will be available in DGL soon.
>
> For the experimental part:
>
> 1) We have added a new experiment in the appendix about walltime to convergence to a target accuracy and it shows that LABOR is faster compared to NS by upto 14% and GraphSAINT by 2 times. We ran this experiment on an A100 GPU with a very large memory bandwidth of 2TB/s which is the main bottleneck for GNN training.
> Running on slower hardware would have made the speed difference even larger because LABOR samples fewer vertices than NS, reducing memory bandwidth requirements to process batches.
> Also, Table 2 includes the iterations per second column, and it shows how fast each step of Figure 1 is.
> With the same batchsize and fanout, LABOR is upto 2.6x faster compared to NS, meaning that it converges upto 2.6x faster than NS on reddit on an NVIDIA T4 GPU.
> Table 2 also shows that LABOR needs upto 13.5x fewer edges compared to LADIES and is more than 3x faster than LADIES overall.
>
> 2) We have added the corresponding accuracy plots to all of our Figures. The accuracy plots and the loss plots pretty much show the same picture.
>
> 3) Our new experiment in the Appendix also compares NS and LABOR to GraphSAINT, LABOR is faster than both NS and GraphSAINT on the same task.
>
> 4) Section 4.2 includes an experiment where sampling methods are evaluated with different batch sizes while sampling the same number of vertices.
> The only hyperparameter of our method is the batch size and the fanouts per layer. Section 4.2 already investigates the effect of changing the batch-size partly.
> We will include an additional experiment for the camera ready version investigating how changing the fanout affects the results.

---

### Author Response · Authors · 2022-11-18
**Explaining the choice of experimental setup, summary of improvements to the paper and our contributions.**

Thanks to the reviewers for their valuable feedback.

We have made the following improvements to the paper:

* We have reran our experiments and updated the plots after fixing a bug related to the Yelp dataset and the LADIES variants.
* We have added the corresponding validation accuracy plots to Figures 1 and 2.
* We have changed the experimental setup in Section 4.2 so that the batch is fixed from the start instead of converging dynamically.
* We have added an additional experiment about wall-time to convergence in Appendix A.4
* Appendix A.3 was improved to include additional discussion about the extension to weighted edge case.

Hamilton et al. (2017) introduced the Neighbor Sampling (NS) algorithm. Since then, numerous studies used NS as their main sampling method while introducing additional improvements. For example, VR-GCN (Chen et al., 2018a) uses NS and caches the historical computed node embeddings. In such studies, one could simply replace NS with LABOR to get performance benefits such as sampling fewer vertices (see Table 2) or using a higher batch-size and/or larger fanout. Liu et al. (2020) and Zhang et al. (2021) make use of NS while using bandits to estimate changing edge weights during training of a GNN model. We leave this as future work but one could again replace NS in their method with LABOR to get similar benefits.

The first layer sampling method was FastGCN (Chen et al., 2018b), and it was superseded by LADIES (Zou et al., (2019)) and AS-GCN (Huang et al., (2018)). The main difference between AS-GCN and LADIES is that AS-GCN uses a neural network to estimate the edge weights so that the extra knowledge of edge weights can further reduce the variance of the sampler, at the expense of the execution time. The same neural network technique can be applied to LABOR (see Appendix A.3 for handling edge weights). Thus, we chose to compare against LADIES from these papers. GNS (Dong et al., 2021) uses a vertex cache and uses NS to sample from this cache first. If enough neighbors do not exist in the cache, then the rest of the neighbors are sampled from the original graph. Again, one can use LABOR instead of NS to get its benefits.

When it comes to subgraph sampling methods, the most relevant studies are Cluster-GCN (Chiang et al., 2019),
GraphSAINT (Zeng et al., 2020), shaDow-GNN (Zeng et al., 2021). GraphSAINT does subgraph sampling in an unbiased
way compared to Cluster-GCN, also their experimental results show GraphSAINT outperforms Cluster-GCN. Despite GraphSAINT
is a different class of sampling algorithm, we included an experiment in the appendix comparing GraphSAINT to NS and LABOR. Our result shows that for a given accuracy target, LABOR and NS are faster than GraphSAINT in terms of walltime. Furthermore, GraphSAINT is a meta sampling method that requires a SAMPLER as input, and they use the input SAMPLER to compute probabilities which they later use to sample vertices or edges. GraphSAINT could also use LABOR as its SAMPLER.
shaDow-GNN requires an EXTRACT method that extracts a subset of the edges in a seed vertex's k-hop neighborhood.
They use NS with a fanout parameter b to extract from the L-hop neighborhood (see their Appendix). One can again replace NS with LABOR with the exact same fanout hyperparameter and leverage benefits of LABOR in comparison to NS.

Another contribution of this paper is to show that Poisson Sampling is preferable to sampling with replacement or sampling without replacement. Both LADIES and AS-GCN have the problem that they are using sampling with replacement in their formulation. In our paper, we show how one can use Poisson Sampling instead and enjoy less variance while still being unbiased compared to LADIES and/or AS-GCN. The benefit from using Poisson sampling is similar to the benefit one gets when one switches from sampling with replacement to sampling without replacement (see Equation 11). To further quantify the reduction in variance, if a vertex has k neighbors sampled via sampling with replacement, its variance is O(1/k) but if one uses Poisson sampling, then the variance is on the order of O(1/k - 1/d) so variance becomes 0 when k is getting close to d.

---

### Comment · Area_Chair_Mz1F · 2022-12-07
**Response to Author Feedback**

Dear Reviewers, thank you so much again for your time on this paper. The discussion phase is still ongoing, how does the author response and other review change your view of the paper?

---

### Decision · Program_Chairs · 2023-01-20

**Decision:**

Reject

**Justification For Why Not Higher Score:**

See above metareview.

**Justification For Why Not Lower Score:**

N/A

**Metareview: Summary, Strengths And Weaknesses:**

While the reviewers appreciated the paper’s organization, clarity, and its hyperparameter analysis, their main concerns were with (a) the baselines compared against, (b) the technical contribution balanced against the overall improvement of the method. Specifically, for (a) the reviewers would have liked to see comparisons against other sampling algorithms (AS-GCN, GNS, GraphSAINT, ClusterGCN). The authors responded by pointing out that AS-GCN and GNS can be combined with their proposed method LABOR (AS-GCN reduces the variance of any sampler, and GNS requires access to a sampler, which could be LABOR). They add an experiment comparing against GraphSAINT and show LABOR converges faster (finally they point out that GraphSAINT improves over ClusterGCN). The issue is with the first two baselines: do AS-GCN and GNS still maintain their improvements when using LABOR instead of NS? Or do they actually harm LABOR’s result, making it worse than NS. This question is left unanswered. For (b), the reviewers point out that the improvements in training loss are marginal, often only observable when zooming into the validation-F1 or training-loss curves. The authors respond pointing out that their method converges more than 2x faster compared to NS and a 3% improvement in accuracy. First, I don’t know which accuracy result they are referring to: in Figure 1, NS often matches/improves over LABOR. Second, this isn’t a direct response to the core point the reviewers are making: the improvements are small and it’s not obvious why. It would be useful if the authors could explain why such improvements are important. As there is a relatively small methodological change made by this paper compared to prior work, the authors should convince us that the results aren’t similar because the change is minor. Given these open points, I believe this work should be rejected at this time. Once these things are clarified this paper will be improved.